# Latency Reduction and Packet Synchronization in Low-Resource Devices Connected by DDS Networks in Autonomous UAVs

**DOI:** 10.3390/s23229269

**Published:** 2023-11-18

**Authors:** Joao Leonardo Silva Cotta, Daniel Agar, Ivan R. Bertaska, John P. Inness, Hector Gutierrez

**Affiliations:** 1Department of Aerospace Engineering, Physics and Space Sciences, Florida Institute of Technology, Melbourne, FL 32901, USA; 2Core Developer, PX4 Autopilot, 8092 Zurich, Switzerland; daniel@agar.ca; 3Control Systems Design and Analysis Branch, NASA Marshall Space Flight Center, Huntsville, AL 35812, USA; ivan.r.bertaska@nasa.gov (I.R.B.); john.p.inness@nasa.gov (J.P.I.); 4Department of Mechanical and Civil Engineering, Florida Institute of Technology, Melbourne, FL 32901, USA; hgutier@fit.edu

**Keywords:** flight control unit, companion computer, DDS network, non-deterministic delay, communication latency, MAVLink, robot operating system, loosely coupled systems

## Abstract

Real-time flight controllers are becoming dependent on general-purpose operating systems, as the modularity and complexity of guidance, navigation, and control systems and algorithms increases. The non-deterministic nature of operating systems creates a critical weakness in the development of motion control systems for robotic platforms due to the random delays introduced by operating systems and communication networks. The high-speed operation and sensitive dynamics of UAVs demand fast and near-deterministic communication between the sensors, companion computer, and flight control unit (FCU) in order to achieve the required performance. In this paper, we present a method to assess communications latency between a companion computer and an RTOS open-source flight controller, which is based on an XRCE-DDS bridge between clients hosted in the low-resource environment and the DDS network used by ROS2. A comparison based on the measured statistics of latency illustrates the advantages of XRCE-DDS compared to the standard communication method based on MAVROS-MAVLink. More importantly, an algorithm to estimate latency offset and clock skew based on an exponential moving average filter is presented, providing a tool for latency estimation and correction that can be used by developers to improve synchronization of processes that rely on timely communication between the FCU and companion computer, such as synchronization of lower-level sensor data at the higher-level layer. This addresses the challenges introduced in GNC applications by the non-deterministic nature of general-purpose operating systems and the inherent limitations of standard flight controller hardware.

## 1. Introduction

The growing interest in Autonomous Aerial and Underwater Vehicles (AAUVs) is rapidly reshaping technologies that support new applications and improve performance. The extensive range of applications and increasing complexity of missions are prompting the development of advanced real-time flight controllers capable of integrating guidance, navigation, and control (GNC) algorithms to reliably execute increasingly complex tasks.

State-of-the-art systems use a divided architecture to address the computational limitations of standard flight control units (FCU), separating tasks between low-level and higher-level layers [1]. This allows computationally intensive GNC functions to be handled by the higher-level layers, which rely on companion computers that are typically more powerful than the low-level FCU. While this division is a good alternative to support mission goals, it introduces a critical drawback in that high-level layers often use general-purpose operating systems known for their non-deterministic time management and unpredictable delays [2]. This creates challenges in developing high performance GNC solutions, as random latencies introduced by the operating system and interface networks can affect the overall performance of motion control systems [1,2,3,4].

The development of autonomous robotic platforms emphasizes software modularity and integration with widely used frameworks. This study presents an innovative method for reducing latency in packet synchronization between a companion computer and a Real-Time Operating System (RTOS) flight controller (a device notably constrained in ternms of computational resources) based on widely available open-source tools: a DDS network in the Robot Operating System 2 (ROS 2) environment, the PX4 FCU, and the eXtremely Resource Constrained Environment-Data Distribution Service (MicroXRCE-DDS) as an intermediary agent between system level layers. Latency measurements for the proposed method are compared against results from the same flight controller software, PX4, with the well-known open-source Micro Air Vehicle Link (MAVLink) protocol, Robotic Operational System (ROS), and MAVROS, a communication bridge between MAVlink and ROS [1,5,6,7,8].

In summary, this paper presents a method to assess latency in communications between the flight controller and companion computer. A comparison based on the measured latency statistics illustrates the advantages of XRCE-DDS compared to standard communication methods based on MAVROS-MAVLink. More importantly, an algorithm to estimate latency offset and clock skew based on an exponential moving average filter is presented, providing a tool for latency estimation and correction that can be used by developers to improve synchronization of processes that rely on timely communication between the FCU and companion computer, such as synchronization of lower-level sensor data at the higher-level layer. This addresses the challenges introduced in GNC applications by the non-deterministic nature of general-purpose operating systems and the inherent limitations of standard flight controller hardware.

## 2. Materials and Methods

### 2.1. Flight Controller Unit

The flight control unit used in this study was the Holybro Pixhawk 4, running a slightly modified PX4 firmware version 1.14.0 [7]. The FCU uses two ARM processors: an STM32F765 as the CPU and an STM32F100 as the IO processor. The internal sensors include an inertial measurement unit (IMU), magnetometer, and barometer, which can be fused with other non-deterministic and deterministic sensors for state estimation during operation. Communication between the FCU and high-level layer was implemented using an FCU UART port at 921600 BPS.

### 2.2. PX4 Firmware

PX4 works through a topic publish/subscribe framework called uORB that uses internal message definitions. This format allows the user to declare custom uORB definitions to be used internally in the FCU, and enables all uORB definitions to be exported for use in an ROS/ROS 2 package. While PX4 can accept MAVlink messages, these have to be converted to uORB in order to properly interface with critical internal functionalities; custom modifications to MAVlink require substantial user effort [7].

### 2.3. ROS/ROS 2

In robotic systems with divided architectures, the two most popular current frameworks are the Robot Operating System (ROS) and Robot Operating System 2 (ROS 2). Although similar in name, their internal publish/subscribe models have fundamentally different protocols. ROS uses a custom message queuing protocol similar to Advanced Message Queuing Protocol (AMQP), in which two types of nodes are used: ROS nodes and a ROS Master. A roscore is a computational entity that can communicate with other nodes and can publish and subscribe to receive and transmit data. The ROS Master acts as central hub for the ROS ecosystem, keeping a registry of active nodes, services, and topics to ease communication between nodes, i.e., discovery registration. Multiple ROS nodes can be executed at the same time and be registered to a roscore; however, cooperation between multiple roscores is not natively supported by ROS [9]. In contrast, ROS 2 implements a decentralized architecture using a Data Distribution Service (DDS), which eliminates the need for a ROS Master and automates the discovery registration. ROS 2 nodes can transmit and receive data without central coordination, with the direct result that ROS2 incurs less latency than ROS. ROS 2 enables the use of several node types (regular, real-time, intra-process, composable, and lifecycle nodes), enabling cooperation between different layers of a system as long as the ROS 2 nodes of interest are under the same DDS domain [10,11].

### 2.4. MAVlink

MAVlink is an open-source messaging protocol widely used by the UAV industry. It uses a header-only message marshaling library optimized for low-resource environments. The implementation of custom messages in MAVlink is a demanding task, in particular when integrating ROS/ROS 2 routines. Custom implementations require changes to the FCU’s source code as well as modifications to the communications bridge used to transmit and translate messages between the ROS/ROS 2 nodes and the FCU [8,10,11,12].

### 2.5. MAVROS

MAVROS is the most commonly used open-source bridge between ROS and MAVlink, converting ROS messages to their MAVlink equivalent. MAVROS source code has to be modified to implement custom MAVlink messages; for instance, a MAVROS extra plugin, which requires a complex set of tasks, is needed to implement a simple addition. Although the MAVROS team has announced their intention to support ROS 2 in the future, the majority of current UAV applications using MAVlink coupled with ROS/ROS 2 use the ROS-compatible MAVROS version. The use of MAVROS and ROS 2 could be implemented using a ROS–ROS2 bridge, although this would increase computational effort and latency, as ROS and ROS2 use different communication architectures [13].

### 2.6. MicroXRCE-DDS

MicroXRCE-DDS (Figure 1) is a protocol that allows low-resource devices to be integrated with a DDS network while maintaining real-time and reliable data distribution capabilities (RTPS). In this investigation, it is used as an ROS 2 middleware to enable nodes running on the FCU to communicate with nodes running on the companion computer [2,14,15].

### 2.7. Companion Computer

The companion is an auxiliary computer running ROS/ROS 2 nodes to support the required mission functionality. It communicates with the FCU using an FTDI cable (USB to UART). For this investigation, the companion computer was an RPi 4 with 8 GB of RAM.

### 2.8. Host Computer

The host was implemented on an x86 computer with an Intel Xeon(R) Gold 6148 CPU running Ubuntu 22.04 with Gazebo 11 and QGroundControl. The host was used as the terminal for the FCU and as a virtual environment to emulate the PX4 internal sensor data [16,17].

### 2.9. Latency Assessment

Latency assessment was implemented via hardware-in-the-loop simulation, and consisted of measuring the time offset between transmitted and received messages on both the FCU side and a high-frequency ROS or ROS 2 node. The forward path consisted of messages sent from the companion computer, simulating data from an external vision sensor to be fused at the PX4’s Extended Kalman Filter (EKF2); latencies were collected when the message was parsed into the estimator module in the FCU. The reverse path consisted of messages sent from the FCU to the ROS or ROS 2 node containing raw measurements from the FCU’s IMU. The choice of messages relates to typical GNC applications such as SLAM or Visual Inertia Odometry, in which rapid IMU feedback and fast state estimate transmission are crucial for mission performance. The messages were custom-modified to carry the same number of 115 bytes.

Figure 2 outlines the PX4’s EKF2 algorithm [18]. The top block shows the main estimator, which uses a ring buffer to account for different sensor sampling frequencies and predict the states in a delayed horizon. The second block is the output predictor, which uses corrected high-frequency IMU measurements for quick state prediction and UAV rate control. Pseudocode and diagrams outlining communications between higher-level and lower-level processors are presented for both scenarios in Algorithms 1 and 2, and in Figure 3 and Figure 4.
**Algorithm 1** Latency Test Node in rospy (ROS)**Class** LatencyTest:   Initialize variables   Initialize publisher and subscriber   Initialize timer **function** Initialize variables    time_packet_creation← empty deque of max length 2    offset_estimated,N,high_dev_counter,high_rtt_counter←0    alpha,beta←0.05    skew←0    convergence_window←500 **function** Initialize publisher and subscriber    Subscribe to ’mavros/imu/data_raw’    Publish to ’mavros/vision_pose/pose’ **function** Initialize timer    Set timer frequency to 200 Hz **function** Sensor callback(msg)    Calculate imu_timestamp,current_time    Obtain imu_time_offset_observed,rtt    Update offset and counters based on rtt and imu_time_offset_observed    Write data to log.txt **function** Update offset and counters(rtt, imu_time_offset_observed)    **if** rtt< 10,000 **then**        Update alpha,beta, and offset using imu_time_offset_observed    **else**        high_rtt_counter←high_rtt_counter+1 **function** Cmdloop callback(event)    Create and publish vio_msg    Append vio_msg.header.stamp to time_packet_creation **function** reset filter    N←0    ${\mathrm{Offset}}_{Estimated}\leftarrow 0$    ${\mathrm{Skew}}_{\mathrm{Estimated}}\leftarrow 0$    $\hat{\alpha }\leftarrow \alpha_{\max}$    $\hat{\beta }\leftarrow \beta_{\max}$    high_deviation_count←0    high_rtt_count←0

**Algorithm 2** Latency Test Node in RCLPY (ROS 2)
**Class** LatencyTest **extends** Node:   Initialize variables   Initialize publisher and subscriber   Initialize timer **function** Initialize variables    time_packet_creation← empty deque of max length 2    offset_estimated,N,high_dev_counter,high_rtt_counter←0    alpha,beta←0.05    skew←0    convergence_window←500 **function** Initialize publisher and subscriber    Create a subscription to ‘/fmu/out/vehicle_imu’    Create a publisher to ‘/fmu/in/vehicle_visual_odometry’ **function** Initialize timer    Set timer frequency to 200 Hz **function** Sensor callback(msg)    Calculate imu_timestamp,current_time    Calculate imu_time_offset_observed,rtt    Update offset and counters based on rtt and imu_time_offset_observed    Write data to “time_offsets.txt” **function** Update offset and counters(rtt, imu_time_offset_observed)    **if** rtt< 10,000 **then**        Update alpha,beta, and offset using imu_time_offset_observed    **else**        high_rtt_counter←high_rtt_counter+1 **function** Cmdloop callback    Create and publish vio_msg    Append vio_msg.timestamp_sample to time_packet_creation **function** reset filter    N←0    ${\mathrm{Offset}}_{Estimated}\leftarrow 0$    ${\mathrm{Skew}}_{\mathrm{Estimated}}\leftarrow 0$    $\hat{\alpha }\leftarrow \alpha_{\max}$    $\hat{\beta }\leftarrow \beta_{\max}$    high_deviation_count←0    high_rtt_count←0


The ROS messages in Table 1 directly translate into the uORB messages shown in Table 2. The red and blue lines in Figure 3 and Figure 4 show the message flow (IMU and VIO) and modules involved. Interpreting the time offset results requires understanding how the FCU firmware and the ROS/ROS 2 nodes estimate the time offset in each message, including how both lower-level and higher-level system time bases can be aligned and how the communication delay is quantified. The time offset is defined as the difference between two clock readings:(1)$${\mathrm{Offset}}_{\mathrm{Observed}}\left(i\right)=\frac{T_{\mathrm{Packet}\;\mathrm{Creation}}\left(i\right)+T_{\mathrm{Current}}\left(i\right)-2\times T_{\mathrm{Remote}\;\mathrm{Stamp}}\left(i\right)}{2}$$
where $T_{\mathrm{Packet}\;\mathrm{Creation}}$ is the time when the packet containing the uORB or MAVlink messages was created, that is, the time when information originated from the ROS/ROS 2 node or FCU was serialized and sent, $T_{\mathrm{Remote}\;\mathrm{Stamp}}$ is the time when the message was received and sent back from the remote level (either the higher-level or lower-level system layer), and $T_{\mathrm{Current}}$ is the current system time. Clock skew is defined as the difference in the register update rate (loop rate) at both the FCU and companion computer. The offset and clock skew are estimated using an exponential moving average filter, as described in [19,20]:(2)$$\begin{array}{cc} {\mathrm{Offset}}_{\mathrm{Estimated}}\left(i\right)= & \alpha \times {\mathrm{Offset}}_{\mathrm{Observed}}\left(i\right)\\ \; & +\left(1-\alpha \right)\times ({\mathrm{Offset}}_{\mathrm{Estimated}}\left(i-1\right)+{\mathrm{Skew}}_{\mathrm{Estimated}}\left(i-1\right)) \end{array}$$
(3)$$\begin{array}{cc} {\mathrm{Skew}}_{\mathrm{Estimated}}\left(i\right)= & \beta \times ({\mathrm{Offset}}_{\mathrm{Estimated}}\left(i\right)-{\mathrm{Offset}}_{\mathrm{Estimated}}\left(i-1\right))\\ \; & +\left(1-\beta \right)\times {\mathrm{Skew}}_{\mathrm{Estimated}}\left(i-1\right) \end{array}$$
where α and β are the filter gains for the offset and skew, respectively. To check convergence of the estimated offset, the message round-trip time is obtained and to determine whether it falls within a maximum threshold:(4)$$T_{\mathrm{RTT}}\left(i\right)=T_{\mathrm{Current}}\left(i\right)-T_{\mathrm{Packet}\;\mathrm{Creation}}\left(i\right)<10\;\mathrm{ms}.$$

If Equation (Equation 4) is true, the deviation between the estimated offset and the latest observed offset is compared against a maximum threshold:(5)$${\mathrm{Offset}}_{\mathrm{Estimated}}\left(i\right)-{\mathrm{Offset}}_{\mathrm{Observed}}\left(i+1\right)<100\;\mathrm{ms}.$$

If Equation (Equation 5) holds, the statistical quality of α and β is assessed for each estimated offset by counting the number of times the expressions used to determine α and β are called in the code:(6)N≥500=ConvergenceWindow⇒Converged.

If Equations (Equation 4) and (Equation 5) hold while Equation (Equation 6) fails (i.e., the number of calls is smaller than the convergence window), then α and β are corrected by interpolation:(7)$$p=1.0-\exp\left(0.5\times \left(1.0-\frac{1.0}{1.0-\frac{N}{500}}\right)\right)$$
(8)$$\hat{\alpha }=p\times \alpha_{\min}+\left(1.0-p\right)\times \alpha_{\max}$$
(9)$$\hat{\beta }=p\times \beta_{\min}+\left(1.0-p\right)\times \beta_{\max}$$
with $\alpha_{\max}$ and $\beta_{\max}$ set as 0.05 and $\alpha_{\min}$ and $\beta_{\min}$ set as 0.003 for convergence of the moving average filter under the tested conditions. The one-way time-synchronized latency is as follows.  
(10)$${\mathrm{Latency}}_{1}=T_{\mathrm{Higher}-\mathrm{level}\Rightarrow \mathrm{Lower}-\mathrm{level}}-{\mathrm{Offset}}_{\mathrm{Higher}-\mathrm{level}\Rightarrow \mathrm{Lower}-\mathrm{level}}$$
(11)$${\mathrm{Latency}}_{2}=T_{\mathrm{Lower}-\mathrm{level}\Rightarrow \mathrm{Higher}-\mathrm{level}}-{\mathrm{Offset}}_{\mathrm{Lower}-\mathrm{level}\Rightarrow \mathrm{Higher}-\mathrm{level}}$$

The respective bounds of 10 ms and 100 ms in Equations (4) and (5) are specific to the implementation in the PX4 platform, and are MAVlink defaults [12,18]. While they can be fine-tuned, this could affect the number of filter resets, in turn impacting estimation of the offset.

### 2.10. Experimental Setup

A hardware-in-the-loop (HIL) setup was used for the latency assessment (Figure 5), based on the following: Pixhawk 4 flight controller FMU-V5 (Lower-level system), companion computer (Higher-level system), and HIL simulation host computer.

## 3. Results

### 3.1. Latency Comparison in the Flight Control Unit

The experiment sequence is described in detail in Section 2.9. After collecting time stamps from the Extended Kalman Filter (EKF2) module, the advantages of DDS network communication in UAVs become quite clear. A reduction in average latency is found; more importantly, the reduction of latency peaks with the use of the MicroXRCE–DDS bridge, which enables more accurate delay prediction for external sensor data fusion. Figure 6 and Figure 7 show results from a stress test at maximum companion computer CPU usage while prioritizing the ROS2 or ROS process in order to check the latency difference when transmitting an external odometry message. In this scenario, the latency is defined as the time elapsed between message creation at the ROS/ROS2 Node and message arrival at the EKF2 module, including the time synchronization process at the FCU, i.e., Algorithm 3 and Equation (Equation 10). Figure 8 shows the consequence of transmitting and receiving high-frequency topics in a low-resource device coupled to a slightly non-robust DDS network due to high companion computer CPU usage, namely, latency peaks, which can usually be mitigated using ROS2 Quality of Service settings [21]. For all assessments using the MicroXRCE-DDS–ROS 2 bridge, the publishers and subscribers used the following configuration settings:Reliability: *BEST_EFFORT*; the publisher attempts to deliver the maximum number of samples possible.History and Queue Size: *KEEP_LAST*; only one message is stored in the processing queue.Durability: *TRANSIENT_LOCAL*; the publishers are responsible for sending the last available message to newly discovered subscribers.
**Algorithm 3** Time Synchronization Algorithm (PX4 uORB/MAVlink [12,18])1:N←02:${\mathrm{Offset}}_{Estimated}\leftarrow 0$3:${\mathrm{Skew}}_{Estimated}\leftarrow 0$4:$\hat{\alpha }\leftarrow \alpha_{\max}$                    ▹ Initialize $\hat{\alpha }$ as $\alpha_{\max}$5:$\hat{\beta }\leftarrow \beta_{\max}$                    ▹ Initialize $\hat{\beta }$ as $\beta_{\max}$6:high_deviation_count←07:high_rtt_count←08: 9:**procedure** update($T_{\mathrm{Current}}$, $T_{\mathrm{Remote}\;\mathrm{Stamp}}$, $T_{\mathrm{Packet}\;\mathrm{Creation}}$)10:    **if** $T_{\mathrm{Remote}\;\mathrm{Stamp}}>0$ **then**11:        ${\mathrm{Offset}}_{Observed}\leftarrow \frac{T_{\mathrm{Packet}\;\mathrm{Creation}}+T_{\mathrm{Current}}-2\times \left(T_{\mathrm{Remote}\;\mathrm{Stamp}}\right)}{2}$12:        $T_{\mathrm{RTT}}\leftarrow T_{\mathrm{Current}}-T_{\mathrm{Packet}\;\mathrm{Creation}}$13:        $\mathrm{deviation}\leftarrow |{\mathrm{Offset}}_{Estimated}-{\mathrm{Offset}}_{Observed}|$14:        **if** $T_{\mathrm{RTT}}<10$ ms **then**15:           **if** est_sync_converged()∧(deviation>100 ms) **then**16:               
high_deviation_count←high_deviation_count+117:               **if** high_deviation_count>5 **then**18:                   reset_filter19:           **else**20:               **if not** est_sync_converged **then**21:                   progress←N/50022:                   $p\leftarrow 1-\exp\left(0.5\times (1-\frac{1}{1-\mathrm{progress}})\right)$23:                   $\hat{\alpha }\leftarrow p\times \alpha_{\min}+\left(1-p\right)\times \alpha_{\max}$24:                   $\hat{\beta }\leftarrow p\times \beta_{\min}+\left(1-p\right)\times \beta_{\max}$25:               **else**26:                   $\hat{\alpha }\leftarrow \alpha_{\min}$27:                   $\hat{\beta }\leftarrow \beta_{\min}$28:               add_sample(${\mathrm{Offset}}_{Observed}$)29:               N←N+130:               high_deviation_count←031:               high_rtt_count←032:        **else**33:           high_rtt_count←high_rtt_count+134: 35:**procedure** add_sample(${\mathrm{Offset}}_{Observed}$)36:    ${\mathrm{Offset}}_{Estimated-1}\leftarrow {\mathrm{Offset}}_{Estimated}$37:    **if** N==0 **then**38:        ${\mathrm{Offset}}_{Estimated}\leftarrow {\mathrm{Offset}}_{Observed}$39:    **else**40:        ${\mathrm{Offset}}_{Estimated}\leftarrow \hat{\alpha }\times {\mathrm{Offset}}_{Observed}+\left(1-\hat{\alpha }\right)\times \left({\mathrm{Offset}}_{Estimated}+{\mathrm{Skew}}_{\mathrm{Estimated}}\right)$41:        ${\mathrm{Skew}}_{\mathrm{Estimated}}\leftarrow \hat{\beta }\times \left({\mathrm{Offset}}_{Estimated}-{\mathrm{Offset}}_{Estimated-1}\right)+\left(1-\hat{\beta }\right)\times {\mathrm{Skew}}_{\mathrm{Estimated}}$42: 43:**procedure** reset_filter44:    N←045:    ${\mathrm{Offset}}_{Estimated}\leftarrow 0$46:    ${\mathrm{Skew}}_{\mathrm{Estimated}}\leftarrow 0$47:    $\hat{\alpha }\leftarrow \alpha_{\max}$48:    $\hat{\beta }\leftarrow \beta_{\max}$49:    high_deviation_count←050:    high_rtt_count←0

### 3.2. Latency Comparison at the Companion Computer

The latency measurements at the companion computer follow the logic shown in Algorithms 1–3. Figure 9 shows the latency values with the elapsed time between an IMU sample and message arrival at the ROS/ROS2 Node, including the time synchronization correction at the companion computer, i.e., the estimated offset from Equation (Equation 2) after convergence is reached. Figure 10 shows the corresponding probability distribution. The messages are time-synchronized at the same node where the latency is assessed, as can be seen by comparing Figure 9 to Figure 6. There is a 349.01 us difference in average latency between the MAVROS–ROS and XRCE–DDS architectures.

### 3.3. Flight Controller CPU and RAM Utilization

The effect of the MicroXRCE-DDS–ROS2 bridge implementation in terms of FCU CPU and RAM usage is significant, and can provide insights into the minimum hardware requirements of future UAV missions. In this study, both MAVROS–ROS and MicroXRCE–DDS bridges were deployed using only those topics listed in Table 1 and Table 2. The CPU demand decreases when using the MicroXRCE-DDS–ROS2 bridge. However, this is related to the capabilities of the chosen FCU hardware (in this case, the PX4 FCU V5, a popular flight control unit) and the number and loop rate of the topics transmitted to the FCU and companion computer. The jump in CPU/RAM usage shown below (Figure 11) corresponds to the beginning of operations, i.e., the start of the Gazebo simulator; a second jump can occur if the MicroXRCE–DDS bridge is not active at the beginning of operations or is not running at its fullest yet, i.e., when the ROS/ROS 2 routines have not yet started.

## 4. Discussion

### 4.1. Latency Reduction and Time Synchronization for Enhanced GNC in UAVs

This study highlights a crucial aspect of software development for complex UAV missions: the inherent latency in multi-layer system architectures. The results presented (Figure 6 and Figure 9) illustrate that the XRCE-DDS–ROS 2 bridge is not only an effective method to reduce communication latency; it is a cornerstone for the development of enhanced GNC algorithms, as the performance of GNC algorithms relies on timely and reliable data exchange [22]. The proposed approach paves the way to develop synchronization corrections for internal and external sensor data within UAV systems, enabling higher performance by providing more accurate timing. The latency correction approach boils down to the estimation of the offset and clock skew, as respectively described in Equations (Equation 2) and (Equation 3). Note that time synchronization is not symmetrical; the latency in each direction needs to be corrected separately.

### 4.2. Time Synchronization Effects in UAV High-Level Layer (Companion Computer)

Synchronization is a fundamental aspect of real-time systems. Lack of accurate timekeeping introduces jitter in control signals and degrades the performance of sensor fusion estimation. The proposed synchronization approach allows ROS or ROS 2 nodes to combine their time-synchronized sensor data (data from a low-resource device) with sensor data at the higher-level layer (e.g., a camera connected to the companion computer), which enables improved sensor fusion at the companion level for operations such as Visual Inertial Odometry (VIO) and SLAM, thereby enhancing loosely coupled distributed UAV systems [3,4,23]. Implementing Algorithm 1 or Algorithm 2 in the high-level layer accounts for asymmetrical network paths and processing delays until data arrival at the GNC ROS/ROS 2 node by using reliable and updated offset and clock skew estimates as opposed to assuming a symmetrical network path with time synchronization at the communication bridge. Furthermore, the proposed implementation accounts for synchronization of the sample time instead of the message header time, i.e., the timestamp when the measurements (e.g., IMU or VIO) were collected, as opposed to the timestamp when the message was sent from the FCU to the companion computer (or vice-versa). A particular case of interest is where $T_{\mathrm{Remote}\;\mathrm{Timestamp}}=T_{\mathrm{Sensor}\;\mathrm{Sample}}$.

### 4.3. Event-Driven Communication between Flight Controller and Companion Computer

The modifications implemented in the PX4 firmware to support this study include a complete event-driven MicroXRCE-DDS–ROS 2 bridge, which allows incoming and outgoing messages to be consumed by requesting processes as soon as they are available. A direct consequence of this is a reduction in the latency’s standard deviation, increasing offset predictability and lowering end-to-end delays.

### 4.4. Trade-Offs of Using DDS Networks in UAV Systems

The decentralized nature of a DDS network improves fault tolerance by improving the system’s resilience against data outliers. Furthermore, robust security features can be employed in DDS (DDS-security). Security enhancements in ROS2 include authentication of nodes joining the DDS domain and encryption of data transmitted through ROS2 topics. DDS enables deployment of companion computers that use real-time operating systems, with potential to further reduce latency and improve synchronization [9,11,24,25].

On the other hand, even when coupled with MicroXRCE-DDS, DDS networks can cause spikes in the FCU’s CPU and RAM utilization if the amount of topics or the message size to be transmitted and received is not properly monitored. This increase is only significant when the FCU publishes and subscribes to a large number of topics (10+), which is not common in UAV missions; nevertheless, it should be carefully monitored, as routine complexity and scalability are related to this issue.

### 4.5. Current Limitations of MAVlink and MAVROS

The MAVlink protocol, although widely used, is expected to become unsuitable in the future as an internal communication protocol between FCUs and companion computers. Message types are constantly evolving as new algorithms, sensors, and data acquisition technologies are developed. Adding a new MAVlink message and streaming it to and from the FCU using MAVROS is a demanding process, and requires extensive knowledge of the MAVlink and MAVROS libraries. ROS depends on a central node that acts as a look-up table with respect to the network nodes, which creates further increases in latency compared to ROS 2, where nodes are discovered automatically. The PX4 FCU firmware uses one type of message for internal communications, uORB, which can be used directly in ROS 2 nodes. On the other hand, MAVROS-ROS uses MAVlink messages that need to be converted to ROS messages on the companion computer side and to uORB in the FCU side.

The results presented here (Table 3) clearly show the increased latency created by the interface layer that converts MAVlink messages to the internal communication protocol in the PX4 FCU. Even with a MAVROS version capable of working in the ROS 2 framework, the delay created by the interface layer remains an issue for future UAV applications. MAVlink messages will continue to be used in several FCU applications, in particular those with well-establish functionalities such as radio transmission of telemetry packets. That said, the MAVlink community should continue to develop alternative communications solutions that can be used with uORB [12].

## 5. Conclusions

Our analysis and testing results show that MicroXRCE-DDS–ROS 2 is a better option as a communication bridge between a high-level companion computer and a low-resource FCU compared to the MAVROS–ROS bridge, having smaller communication latency and providing operation closer to real-time in GNC applications. The decentralized nature of DDS networks enables enhanced security features and risk reduction in AAUV missions. The proposed approach to time synchronization and latency correction improves performance in multi-layered AAUV systems, as it allows proper time alignment of sensor data from lower-level layers; algorithms in the higher-level layer have have access to data with more accurate timestamps. This can be particularly beneficial in sensor fusion for depth and visual-inertial odometry applications, where IMU measurements from the FCU need to be time-synced with camera frames at the companion computer for improved performance. An algorithm to estimate latency offset and clock skew based on an exponential moving average filter has been presented, providing a tool for latency estimation and correction that can be used by developers to improve synchronization of processes that rely on timely communication between an FCU and companion computer, such as the synchronization of lower-level sensor data at the higher-level layer. This addresses the challenges introduced in GNC applications by the non-deterministic nature of general-purpose operating systems and the inherent limitations of standard FCU hardware.

### Future Work

Assessment of the effect of high-level latency correction in the performance of GNC algorithms, in particular motion control.Assessment of scalability effects on latency and latency correction when using denser ROS 2 routines, i.e., when more DDS topics are shared between the FCU and companion computer.A latency comparison between the MicroXRCE-DDS–ROS 2 bridge and other emerging technologies, such as Zenoh-Pico–ROS 2 [26].

## Figures and Tables

**Figure 1 sensors-23-09269-f001:**
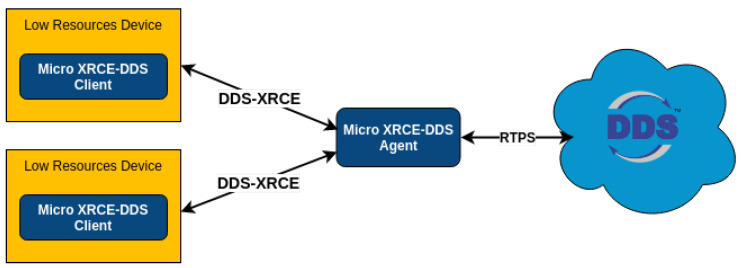
MicroXRCE-DDS architecture [15].

**Figure 2 sensors-23-09269-f002:**
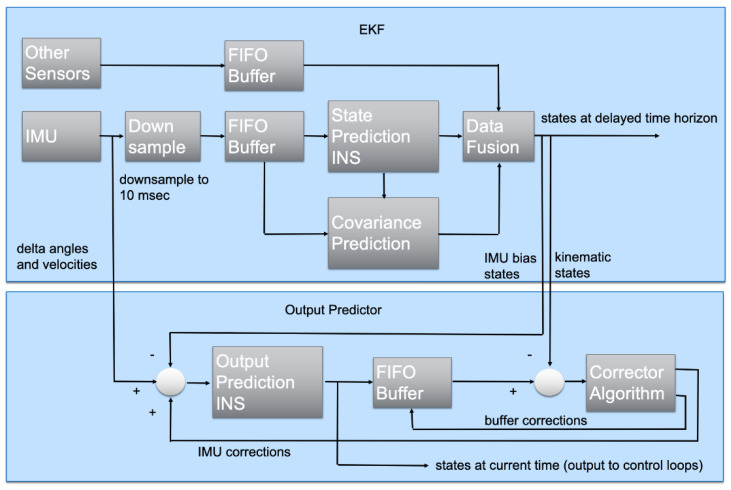
PX4 EKF2 architecture [18].

**Figure 3 sensors-23-09269-f003:**
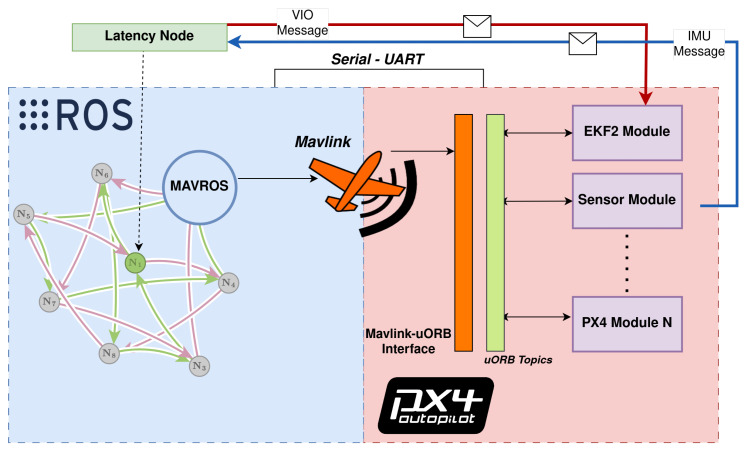
ROS and FCU communications for latency assessment using MAVROS/MAVlink bridge.

**Figure 4 sensors-23-09269-f004:**
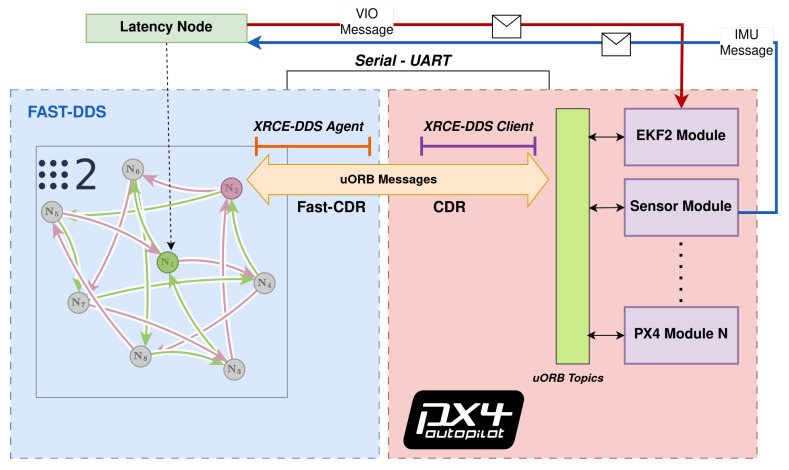
ROS 2 and FCU communications for latency assessment using the MicroXRCE–DDS bridge.

**Figure 5 sensors-23-09269-f005:**
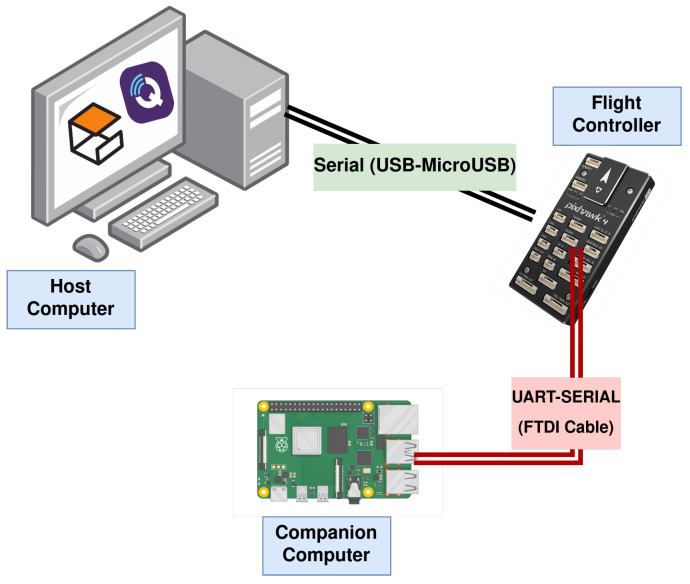
Hardware-in-the-loop setup for end-to-end latency measurements.

**Figure 6 sensors-23-09269-f006:**
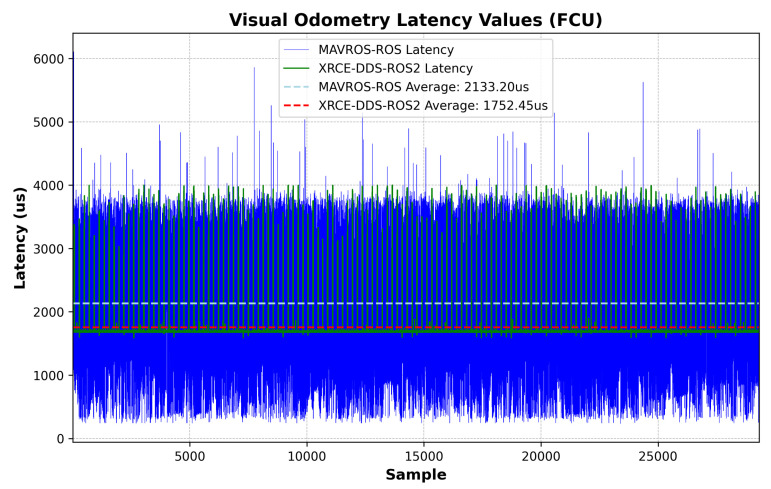
Visual odometry message: latency comparison, FCU.

**Figure 7 sensors-23-09269-f007:**
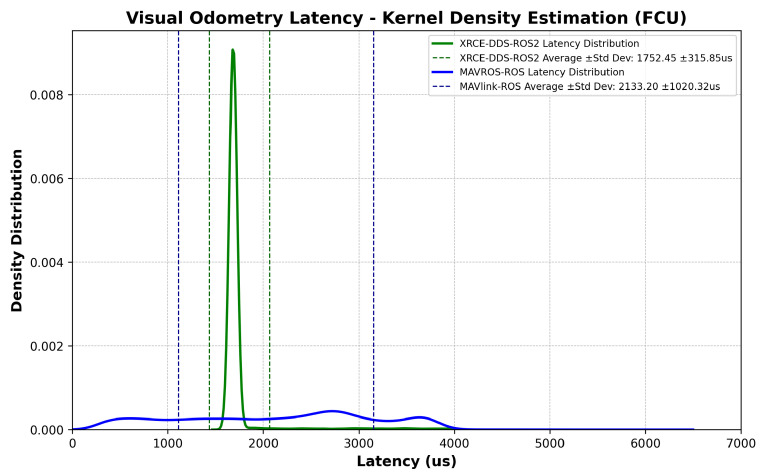
Visual odometry message: comparison of latency probability distribution, FCU.

**Figure 8 sensors-23-09269-f008:**
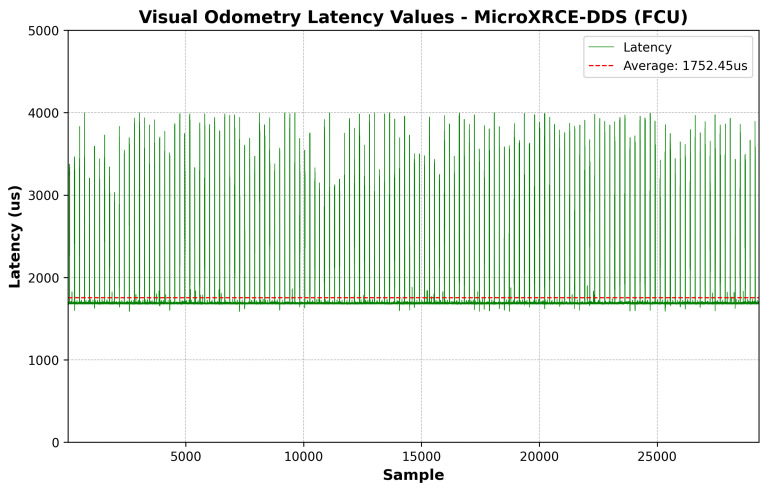
Visual odometry message: latency when using MicroXRCE-DDS–ROS 2.

**Figure 9 sensors-23-09269-f009:**
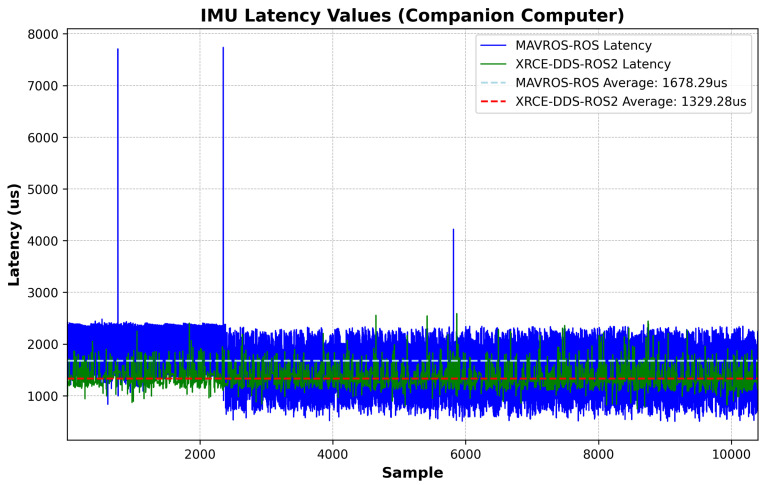
IMU message: latency comparison, companion computer.

**Figure 10 sensors-23-09269-f010:**
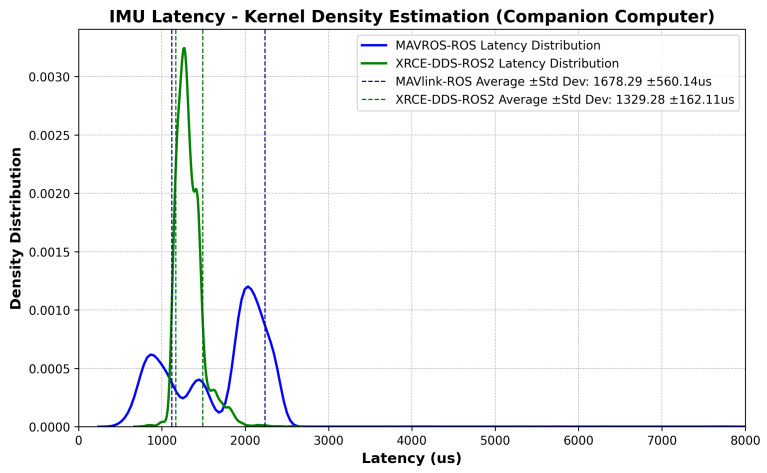
IMU message: comparison of latency probability distribution, companion computer.

**Figure 11 sensors-23-09269-f011:**
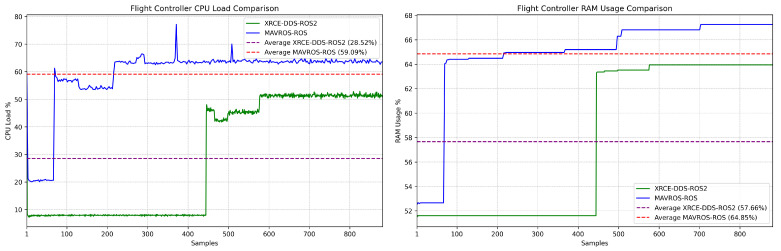
Pixhawk 4 FMU-V5: CPU and RAM load comparison during latency testing.

**Table 1 sensors-23-09269-t001:** ROS: FCU communication using MAVROS bridge.

MAVROS Topic	ROS Message	Rate (Hz)
mavros/vision_pose/pose ^1^	PoseStamped	200
mavros/imu/data_raw ^2^	Imu	200

^1^ Published from companion computer, assessed at FCU. ^2^ Published from FCU, assessed at companion computer. The message and topic rate were custom-modified for comparison; the original rate was 50 Hz.

**Table 2 sensors-23-09269-t002:** ROS 2: FCU communication using MicroXRCE-DDS bridge.

Topic	uORB	Rate (Hz)
vehicle_visual_odometry ^1^	VehicleOdometry	200
vehicle_imu ^2^	VehicleImu	200

^1^ Published from companion computer, assessed at FCU. ^2^ Published from FCU, assessed at companion computer. Custom definition.

**Table 3 sensors-23-09269-t003:** Results summary.

Method	Path	FCU Resource Utilization (%) ^1^	Average Latency and Standard Deviation (Microsec)
MicroXRCE-DDS–ROS 2	Companion Computer → Flight Controller	CPU: 28.5	1329 ± 162
	Flight Controller → Companion Computer	RAM: 57.6	1752 ± 315
MAVlink–MAVROS (ROS)	Companion Computer → Flight Controller	CPU: 59.1	1678 ± 560
	Flight Controller → Companion Computer	RAM: 64.9	2133 ± 1020

^1^ Average values, Holybro Pixhawk 4 FMU-v5.

## Data Availability

Data and codes are available upon reasonable request.

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
