# Peer review of "Latency Reduction and Packet Synchronization in Low-Resource Devices Connected by DDS Networks in Autonomous UAVs"

_sensors, 2023, doi:10.3390/s23229269_

Round 1
Reviewer 1 Report
Comments and Suggestions for Authors
Dear authors;
1-Can you summarize in a few sentences what all this work is about at the end of the introduction?
2-Theabstract needs more interest and rewriting of some paragraphs.
3-There are still some aspects that can be improved (for grammar and punctuation). Improve the technical writing of your paper, where there are several grammatical errors and spelling I think they need to be checked out.
4-The conclusion needs more effort to elaborate on the achieved results with respect to future work,
5-There are still some aspects regarding the obtained results discussions that are missing. Can you please address your achievements well?
6-The practical part is very important, ,
7-Future work is an important part of the conclusion.
8-The results are still not matured well. Can you clearly mention what your objectives clearly for each result you got during the discussion process?
I loved this work and I feel it is very good. I hope these comments will help you improve this work after a major revision.
Regards
Comments on the Quality of English LanguageSome sentences are too long and dificult to understand. Can you reduce the length and fix the languge.
Author Response
Thanks for your comments. Please see attached file for detailed reply to your comments.

Reviewer 2 Report
Comments and Suggestions for Authors
Please see the attached

Moderate editing of English language required
Author Response

(The authors gave the same response as above.)

Reviewer 3 Report
Comments and Suggestions for Authors
I have read your manuscript .I known your proposed method aims at enhancing the system performance.But there are some problems you should revise.
1. The reference document should include more related and recent articles.
2. The section 4.6 "future work"should be palced in Section of Conclusion.
3.The Conclusion should include more numeric information in detail to support your results.
4.The experimental results should consider the more practical working condition,which can illusrate reliability of your proposed method.
5.A few of English grammar mistakes should be corrected
Comments on the Quality of English Language
A few of English gramar mistakes should be corrected.
Author Response

(The authors gave the same response as above.)

Reviewer 4 Report
Comments and Suggestions for Authors
Please see the attached pdf file.

Minor editing of English language required
Author Response

(The authors gave the same response as above.)

Round 2
Reviewer 1 Report
Comments and Suggestions for Authors
My decision to accept the paper in present form.
Comments on the Quality of English LanguageThe English structure needs to be improved
Author Response
Dear Reviewer,
Based on your second list of comments, your first list of comments, and my response to your previous comments, it seems that your observations have already been addressed in the latest version of the paper.
Thanks for your consideration - best regards.
Reviewer 2 Report
Comments and Suggestions for Authors
Accept in present form
Comments on the Quality of English LanguageMinor editing of English language required
Author Response

(The authors gave the same response as above.)

Reviewer 3 Report
Comments and Suggestions for Authors
The discussion and disccusion still should be written in more detailed.
Comments on the Quality of English LanguageThere are some English language problme needed to be corrected.
Author Response

(The authors gave the same response as above.)

Reviewer 4 Report
Comments and Suggestions for Authors
Please see the attached pdf file.

Minor editing of English language required
Author Response

(The authors gave the same response as above.)
